# Adaptive and Robust Operation with Active Fuzzy Harvester under Nonstationary and Random Disturbance Conditions

**DOI:** 10.3390/s21113913

**Published:** 2021-06-06

**Authors:** Yushin Hara, Keisuke Otsuka, Kanjuro Makihara

**Affiliations:** Department of Aerospace Engineering, Tohoku University, Sendai 980-8579, Japan; yushin.hara.p1@dc.tohoku.ac.jp (Y.H.); keisuke.otsuka.d6@tohoku.ac.jp (K.O.)

**Keywords:** wideband vibration, nonlinear harvesting, self-adaptive control, semi-active control

## Abstract

The objective of this paper is to amplify the output voltage magnitude from a piezoelectric vibration energy harvester under nonstationary and broadband vibration conditions. Improving the transferred energy, which is converted from mechanical energy to electrical energy through a piezoelectric transducer, achieved a high output voltage and effective harvesting. A threshold-based switching strategy is used to improve the total transferred energy with consideration of the signs and amplitudes of the electromechanical conditions of the harvester. A time-invariant threshold cannot accomplish effective harvesting under nonstationary vibration conditions because the assessment criterion for desirable control changes in accordance with the disturbance scale. To solve this problem, we developed a switching strategy for the active harvester, namely, adaptive switching considering vibration suppression-threshold strategy. The strategy adopts a tuning algorithm for the time-varying threshold and implements appropriate intermittent switching without pre-tuning by means of the fuzzy control theory. We evaluated the proposed strategy under three realistic vibration conditions: a frequency sweep, a change in the number of dominant frequencies, and wideband frequency vibration. Experimental comparisons were conducted with existing strategies, which consider only the signs of the harvester electromechanical conditions. The results confirm that the presented strategy achieves a greater output voltage than the existing strategies under all nonstationary vibration conditions. The average amplification rate of output voltage for the proposed strategy is 203% compared with the output voltage by noncontrolled harvesting.

## 1. Introduction

Vibration energy harvesting technology has attracted considerable research attention because of its role as a power supply for electronic devices. This technology provides various smart applications depending on the vibration scale and harvester size, for example, power supply to electric vehicles with magnetic spring suspensions from rough road driving [1], or active limb prosthetics for walking motion [2]. A self-sensing harvester, which measures temperature [3], displacement [4], and body motions [5] without the use of specific sensors and batteries, has also been proposed for advanced applications. Transducers for vibration energy harvesting include tuned mass dampers [6,7], magneto-rheological elastomers [8], electromagnetic devices [9,10], electrets [11,12], magnetostrictive elements [13,14,15], and piezoelectric elements [16]. Piezoelectric transducers convert mechanical energy into electrical energy and vice versa as direct and inverse piezoelectric effects [17,18]. This study focuses on piezoelectric vibration energy harvesting (PVEH). Important objectives of PVEH studies are to design a method that can output voltage efficiently from disturbances at identical scales [19] and a method that achieves effective harvesting under the vibration in real environments such as aerodynamic excited vibrations [20,21]. Introducing switch elements to a harvesting circuit and actively switching the circuit connections can improve the harvesting performance [22]. Therefore, switch control strategies that feedback the state values of the harvester were utilized to address efficient harvesting. An energy harvesting circuit comprising switch elements is called an active harvester. To realize effective harvesting, a charge inversion circuit is used to perform the primary function of the active harvester. A charge inversion circuit proposed by Richard et al. [23] inverts polarities of the stored electric charge in the piezoelectric transducer and the corresponding piezoelectric voltage by operating the circuit-housed switch element. This method is discussed from the viewpoint of the energy analysis [24,25], the transition response including control analysis [26], and the feasibility of implementation [27]. With this function, mechanical energy can be converted into electrical energy more effectively from the vibrating structure to the electric circuit than under a noncontrolled scenario [28,29,30]. The strategy of the charge inversion circuit ensures the matching of the polarities of the vibrating structure velocity and piezoelectric voltage [31]. A charge inversion circuit that is used for semi-active vibration control can achieve advanced vibration suppression performance compared to passive control. It demonstrates an effective energy-saving capability compared to a fully active one. The advanced charge inversion circuit operation with specific control strategies such as the bang-bang control-based strategy [32] has been proposed. Guyomar et al. [33] mounted a charge inversion circuit on the PVEH. This method has been widely used as a PVEH because of its efficient harvesting performance. Richard et al. [23] used synchronized switching damping on an inductor (SSDI) as a vibration control strategy for the charge inversion circuit. Guyomar et al. [33] proposed synchronized switch harvesting on inductor (SSHI), in which the SSDI strategy was renamed as a control strategy for harvesting. The SSHI strategy requires only the function of detecting the peaks of mechanical vibrations. The peak detection considers the signs of the harvester electromechanical conditions but not the amplitudes. Therefore, the switching strategy should be improved to design functions that consider not only the signs but also the amplitudes of the harvester conditions.

To fulfill the aforementioned requirement, control-engineering-based switching strategies were proposed with a self-powered digital controller [34,35,36]. Yamamoto et al. [36] proposed a control strategy wherein the linear-quadratic regulator theory was introduced as a control strategy for the charge inversion circuit without relying on peak detection. This strategy controls the charge inversion circuit to match the polarity of the piezoelectric charge and that of the optimal input trajectory determined by the linear-quadratic regulator. Yoshimizu et al. [37] and Makihara et al. [38] proposed switching strategies that drive the charge inversion circuit intermittently at some extremes, but not at all extremes, thereby accomplishing intermittent switching. This strategy for intermittent switching action is termed switching considering vibration suppression (SCVS). These strategies perform sparser switching actions compared to those based on the original SSHI strategy. Pause durations between each switching action provide a benefit in terms of mechanical vibration recovery for efficient harvesting. The suppressed mechanical vibration amplitude caused by the switching action is recovered by the input energy from the disturbance until nonswitching durations. If the harvester is evaluated based on the magnitude of the final attained energy instead of the amount of instantaneous work, an efficient harvester needs to ensure large piezoelectric charge amplitudes and large mechanical vibration amplitudes, thereby achieving high output voltage amplitudes. An active harvester has two vibration-damping roles: passive damping and active damping. Passive damping occurs from a dashpot element in the mechanical components and a resistor in the electrical components. Passive damping suppresses the mechanical vibration with energy dissipation on these components. This study does not discuss the passive damping effect. On the other hand, active damping occurs from the actuation mechanism of the piezoelectric transducer, which is an energy transformer in the harvester. Active damping suppresses the mechanical vibration with the piezoelectric charge flipping by the charge inversion circuit. The mechanical vibration damped by active damping leads to undesirable switching actions and harvesting performance degradations.

Threshold-based strategies that employ a threshold for considering the amplitude of the harvester state values have been proposed. The previously proposed threshold-based strategies are categorized as mechanical strategies and electrical strategies. A harvester adopting mechanical-threshold-based strategy has two mechanical switch elements mounted on the top and bottom of the harvester cantilever beam. The distance between the mechanical switches and the stational position of the cantilever beam acts as the threshold. Switching actions are performed only when these mechanical switches are pushed by the vibrating cantilever, and they allow effective harvesting because the large magnitude of the displacement is always satisfied. Liu et al. [39], Liu et al. [40,41], Giusa et al. [42], and Asanuma et al. [43] used the mechanical vibration amplitude as the threshold. The electrical-threshold-based strategy provides the reference voltage to the comparator mounted on the harvesting circuit. The comparator performs the switch element operation only when the magnitude of the piezoelectric voltage is larger than the reference voltage. Lallart [44] proposed a control strategy named SSHI-threshold (SSHI-t) for a charge inversion circuit under multimodal vibration conditions to avoid undesirable switching actions using a threshold. The SCVS strategy [37,38] avoids the reduction in the piezoelectric charge caused by switching, which is similar to threshold-based strategies because all switching actions are performed at large mechanical vibration amplitudes caused by vibration recovery during the pause duration.

If the disturbance working on the harvester is predictable and stationary, the threshold can be adjusted to an appropriate magnitude during the harvester design phase. The nature-based disturbance is not predictable and changes its scale over time. The mechanical vibration excited on the harvester by the disturbance becomes nonstationary during the harvesting process [45,46,47]. Because the assessment criterion for desirable switching depends on the scale of displacement amplitude, the fixed threshold has low robustness to the operation of the harvester in a real environment. Therefore, the tuning mechanism must be introduced to the time-varying threshold. In previous research, a shape-memory alloy, such as the tuning mechanism of the mechanical threshold [40,41], as well as an output voltage of the time-varying electrical threshold [44], were presented. These improved harvesters provide effective output under complex vibration conditions. In contrast, the adaptive SCVS (ASCVS) strategy without a specific physical threshold [37] estimates both electromechanical variables and evaluates the execution of the switching action at all vibration peaks with the digital calculation.

This study modifies the threshold using a control engineering approach. Simple feedback control with a specific reference cannot be used with the threshold tuning method because the disturbance measurement, which is necessary for the reference derivation, is computationally expensive. A tuning algorithm needs to tune an appropriate threshold from the limited information without a reference. Fuzzy control theory accomplishes appropriate control under a scenario with impossible mathematical discussions [48]. Fuzzy control theory requires experience-based linguistic control logic to achieve adequate control. Even if the causal relationship between the observables and manipulated values is not mathematically described, the manipulated value can be determined from the ambiguous causal relationship. This feature of fuzzy control theory is appropriate for a tuning algorithm of the deriving threshold with no physical dimension from limited information. The ability of fuzzy control to deal with ambiguity is directly related to improving the adaptability and robustness of effective harvesting.

In previous research, we developed a threshold-based ASCVS (ASCVS-t) strategy that enhances output voltage using fuzzy control theory [49]. The investigation into the ASCVS-t strategy in steady harmonic vibrations suggested that it would have excellent performance in harvesting. This paper had two novel objectives compared to our previous report [49]. The first was to confirm the operation performance of the proposed strategy under more realistic disturbances than the stationary and deterministic disturbances that the active harvester was previously exposed to. Rantz and Roundy [19] reported on the features of disturbances that serve as sources of vibration energy for the harvester. In vibration environments excluding the machine source disturbance, the features of the disturbance were mainly unsteady disturbance, filtered noise, and white noise. The evaluation and discussion of the operation of the harvester under nonstationary and random disturbances [50,51] have not been carried out for the ASCVS-t strategy. The second objective is to validate the advantage of the ASCVS-t strategy explicitly. The ASCVS-t strategy enables output voltage amplification without off-line tuning by manual interference in various disturbance environments. When the disturbances are measurable and known, modeling the disturbance in advance allows for effective control and provides a high output voltage based on optimal control theory [36]. Additionally, it was reported that excessive control leads to output voltage attenuation owing to vibration suppression. The intermittent control can recover the suppressed mechanical vibration to the forced excitation response amplitude under the steady disturbance environment [37]. However, strategies except the ASCVS-t strategy cannot achieve effective harvesting under nonstationary and stochastic disturbances because such strategies address neither the unknown disturbance modeling nor the large vibration amplitude maintenance by intermittent switching against the nonstationary disturbance. The ASCVS-t strategy is equipped with a fuzzy control theory that can ambiguously treat physical phenomena. The ASCVS-t strategy has the potential to maintain a high output voltage without disturbance modeling owing to fuzzy modeling. Moreover, the vague decision of the ASCVS-t strategy can alternately change the control style between intermittent and continuous control depending on the disturbance and the harvester conditions. In the previous paper on the ASCVS-t strategy, the advantages of the ASCVS-t strategy were not sufficiently presented because the experiments were conducted under a steady disturbance environment. In accordance with the above two objectives, this paper presents the harvesting experiments using the ASCVS-t strategy under nonstationary disturbance conditions and discusses the aforementioned advantages of the strategy. We first present the advantages of threshold-based strategies that consider the polarities and amplitudes of the electromechanical state values of the harvester and outline the ASCVS-t strategies. Then, we compare the ASCVS-t strategy with existing strategies. We present the results of the assessment experiment under nonstationary and random vibration conditions. The output voltage of the harvester with the ASCVS-t strategy is greater than that of the other strategies.

## 2. Materials and Methods

### 2.1. Harvester Model

A harvester comprises three elements: a transducer, and mechanical and electrical components. The mechanical component of the harvester is modeled as a two-degree-of-freedom (2-DOF) structure with a single piezoelectric transducer inserted between the first mass and the base (Figure 1). The 2-DOF structure possessing two resonance frequencies provides broadband harvesting more than a 1-DOF structure against the broadband disturbance [52,53,54]. When the transducer is inserted between Mass 1 and Mass 2, the wirings connecting the transducer to the harvesting circuit are deformed along with the structural vibration. To avoid fatigue rupture of the wirings, the transducer is inserted between the fixed base and Mass 1. The role of damping in the harvester model is to accurately calculate output voltage from numerical analysis used for the proposed control strategy. The mechanical energy in the harvester without damping excited at the resonance frequencies keeps increasing infinitely, which does not emulate real harvester dynamics. Such excitation conditions were considered in this paper. Damping was introduced to prevent discrepancies between experiments and numerical analysis and to appropriately design the proposed control strategy. The positive influence on the stability of the introduced damping becomes notably clear when the disturbance frequency corresponds to the resonant frequencies of the harvester. The proposed strategy feeds back both amplitudes of voltage and vibration displacement. The magnitude of these values clearly depends on the damping at the resonant frequencies. The diverged values cause the proposed strategy to be unstable. The introduction of damping in the harvester model is necessary to avoid instability in the harvester control. The right side of Figure 1 shows the harvesting circuit composed of two configurations. The first configuration is the standard harvesting composed of a piezoelectric transducer, a full-bridge rectifier, and a smoothing capacitor *C*_s_. The second configuration is the charge inversion circuit composed of a switch device *S* and an inductor *L*. *R_L_* is the parasite resistance in the inductor *L*. The harvester circuit has three circuit equations based on the conditions of the two semiconductor elements, a switch element, and a rectifier, thereby dominating the circuit connection. The switch element changes its status in accordance with the control signals determined by the switching strategy. The rectifier used in this paper is passive and conducts both input and output circuits only when the amplitude of the input AC voltage is larger than that of the output DC voltage.

To describe all dynamics of the harvester, the state values are determined as
(1)$$\vec{z}\equiv {\left[\begin{array}{ccccccc} x_{1} & x_{2} & {\dot{x}}_{1} & {\dot{x}}_{2} & Q_{p} & {\dot{Q}}_{p} & Q_{s} \end{array}\right]}^{T}$$
where *x*, *Q*_p_, and *Q*_s_ denote the mass displacement, the electric charge stored in the piezoelectric transducer, and that stored in the smoothing capacitor, respectively. The governing equations of electric charges depend on the circuit connections (Figure 2). The electromechanical coupling equation of the harvester in the state space expression is expressed as
(2)$$\frac{d}{dt}\vec{z}=\left[\begin{array}{ccc} A_{\mathrm{mech}} & A_{e\to m} & 0\\ A_{m\to e} & A_{\mathrm{elec}} & 0\\ A_{m\to o} & 0 & A_{\mathrm{output}} \end{array}\right]\vec{z}+\left[\begin{array}{c} B_{\mathrm{mech}}\\ 0\\ 0 \end{array}\right]f_{\mathrm{dist}}$$
where *f*_dist_ denotes the disturbance. The elements in each block matrix are described in Appendix A. Each component in the electromechanical model with a piezoelectric transducer used in this paper employs the following assumptions:The mechanical components are composed of two masses connected in series by two springs. This system is modeled as a linear 2-DOF structure and has two vibration modes.The piezoelectric transducer mechanically deforms only in one direction. Both direct and inverse piezoelectric effects are discussed for a range of small deformations and do not consider hysteresis properties. The transducer is installed between the first vibrating mass and the fixed base. The deformation of the transducer corresponds to the displacement of the first mass.The electrical components are modeled using only passive components. Semiconductor elements in the electric circuit are modeled linearly as either open or closed electrical conditions. Because the forward voltage in diode elements is sufficiently small compared to the piezoelectric voltage, the forward voltage of the diodes is neglected.

### 2.2. Mechanism of Charge Inversion Circuit and Switching Action

The charge inversion circuit is used to flip the polarity of the piezoelectric charge to maintain the positive instantaneous work of the piezoelectric transducer from the mechanical to electrical parts. The polarity flip is also used to increase the amplitude of the piezoelectric charge compared to its amplitude before a switch control action. The transferred energy per unit time from the mechanical component to the electrical component is improved by the accurate charge inversion circuit control. The mechanism of the charge inversion circuit is described below. The discussion is valid only when the *LC* electrical vibration frequency is considerably larger than the mechanical vibration frequency. The switch device changes from an open state to a closed state. Subsequently, it changes its state again after half of the *LC* series resonance period. The sequence of switch state changes is called a switching action, and the analytic solution of this phenomenon is given as
(3)$$Q\left(\frac{\pi }{\omega_{e}}\right)=\gamma \left(Q_{\mathrm{before}}-b_{p}C_{p}x_{s}\right)\left\{\cos\left(\sqrt{1-\zeta_{e}^{2}}\pi \right)+\frac{\zeta_{e}}{\sqrt{1-\zeta_{e}^{2}}}\sin\left(\sqrt{1-\zeta_{e}^{2}}\pi \right)\right\}+b_{p}C_{p}x_{s},$$
where
(4)$$\omega_{e}\equiv \frac{1}{\sqrt{LC_{p}}},\;\;\;\;\;\zeta_{e}\equiv \frac{R_{L}}{2}\sqrt{\frac{C_{p}}{L}},\;\;\;\;\;\gamma \equiv \exp\left(-\frac{\pi R_{L}}{2}\sqrt{\frac{C_{p}}{L}}\right),\;\;\;\;\;Q_{\mathrm{before}}\equiv Q\left(0\right),$$
where *x*_s_ denotes a constant value of the displacement when the switch element changes from open to closed. Although the displacement is a variable of time, this variable is handled as the constant while the switch element is closed owing to the assumption regarding the magnitude relation of both the mechanical and electrical vibration frequencies. Here, we assume $\zeta_{e}\ll 1$. The resulting electric charge *Q*_after_ is expressed as
(5)$$Q_{\mathrm{after}}=-\gamma Q_{\mathrm{before}}+\left(1+\gamma \right)b_{p}C_{p}x_{s}$$

Indicating that the polarity of the piezoelectric charge becomes the opposite, and its amplitude is amplified compared to that before switching actions only when the relationship between *Q*_before_ and *x*_s_ satisfies the following inequality:(6)$$x_{s}>\Gamma_{\mathrm{criterion}}Q_{\mathrm{before}}$$
where
(7)$$\Gamma_{\mathrm{criterion}}\equiv -\frac{\left(1-\gamma \right)}{\left(1+\gamma \right)b_{p}C_{p}}$$

As the piezoelectric voltage is increased by the piezoelectric charge, the piezoelectric transducer with a large piezoelectric voltage caused by switching actions can supply more energy compared to standard harvesting (STDH). Inequality (6) presents two suggestions for determining the switching moment to increase the amount of the piezoelectric charge:A switching action should be performed when the signs between the displacement and charge are opposite.A switching action should be performed when the magnitude of displacement is sufficiently larger than the right term depending on the piezoelectric charge in inequality (6).

When the mechanical vibration in the harvester has only one dominant frequency, implementing switching actions at the peaks of the mechanical displacement satisfies the aforementioned two suggestions. Moreover, these moments provide the largest increment of the piezoelectric charge at one switching action because the peak corresponds to the maximal magnitude of the mechanical vibration. The SSHI strategy advocated at the beginning of the active harvester development includes peak switching as an appropriate moment of switching actions.

### 2.3. Threshold-Based Switching Strategy for Complex Vibration

Disturbances in real-world environments lead to complex vibrations that have two or more dominant frequencies and nonstationary waveforms on the harvester. The original SSHI strategy cannot handle such complex vibrations because all peaks of the mechanical displacement do not always satisfy inequality (6). The switching strategy must be able to determine the magnitude of the relationship between both electromechanical variables to effectively operate the harvester following inequality (6) in a real-world environment.

The time-invariant threshold is available only when the disturbance vibrating the harvester in the operating environment is predictable and stationary. The time-varying threshold and its tuning mechanism achieve effective harvesting under unpredictable and nonstationary disturbance conditions during the harvesting process. The simplest adaptive time-varying threshold uses the root-mean-square (RMS) value of the piezoelectric voltage or mechanical displacement. Even though the level of disturbance changes during the harvesting process, the RMS values can follow the change in those magnitudes. The equation for RMS calculation is defined as
(8)$$X_{\mathrm{RMS}}\left[k\right]\equiv {\left(\frac{1}{N_{1}}\sum_{\tau =0}^{N_{1}}X^{2}\left[k-\tau \right]\right)}^{1/2}{}_{,}$$
where
(9)$$X\equiv {\left[\begin{array}{cc} V_{p} & x_{1} \end{array}\right]}^{T},\;\;\;N_{1}\equiv T_{1}/T_{s}$$
where *T*_1_, *T*_s_, and *k* denote the higher resonance period of the two vibrating masses, sampling period, and discrete time, respectively. The RMS values are used with the previous state values to indirectly detect the trend of changes in the present condition of the harvester. When a short resonant period is used for RMS calculation, a small change in the harvester conditions is strongly reflected in the calculation result. Small changes in the harvester are undesirable to tune the time-varying threshold because it induces control chatter; therefore, the longer resonance period is adopted. In this research, the piezoelectric voltage-based threshold is adopted as a criterion for switching execution. Hence, the threshold is
(10)$$V_{\mathrm{th}}\left[k\right]\equiv \beta \cdot V_{\mathrm{RMS}}\left[k\right]$$
where *V*th denotes the threshold and *β* represents the threshold coefficient. Switching actions execute only when the piezoelectric voltage amplitude is higher than the threshold.

The magnitude of the threshold coefficient determines the dominant harvesting actions. The harvesting actions are roughly separated into three control types: STDH, SSHI, and SCVS. Figure 3 shows the time history samples of the piezoelectric charge, and the piezoelectric voltage and the threshold under each control type in the steady state.

The STDH control type, which occurs by the exceedingly large threshold coefficient magnitude, does not perform switching actions in the harvesting process. This type relinquishes the opportunities of the increment of the piezoelectric charge by switching actions and should be avoided by decreasing the threshold magnitude.The SSHI control type, which occurs because of the exceedingly small threshold coefficient magnitude, performs switching actions, including both desirable and undesirable actions. This type leads to the attenuation of the piezoelectric charge because of the undesirable switching actions and should be avoided by increasing the threshold magnitude.The SCVS control type, which occurs because of the adequate threshold coefficient magnitude, accomplishes appropriate intermittent switching actions for effective harvesting. The present threshold design is aimed at this type.

The key technology of the tuning algorithm for the time-varying coefficient threshold is based on the detection of the present control type of the harvester. The control type does not relate to the magnitude of the electromechanical variables of the harvester under unknown vibration conditions. Monitoring changes in the magnitude of both electromechanical variables during the harvesting process provides rough inferences of the present control type of the harvester indirectly. However, the calculation of the concrete manipulated value of the threshold coefficient from the vague inference is unrealistic under general mathematical discussions. The fuzzy control theory in a threshold tuning algorithm was used to detect the present control type of the active harvester that has nonstationary vibrations. In addition, it was employed to calculate the concrete manipulated value of the threshold from a vague inference to maintain the SCVS control type. A detailed explanation of the threshold coefficient tuning algorithm was provided in our previous publication [49]. The summary of the proposed algorithm is described as follows: The controller calculates the RMS values of mechanical displacement and piezoelectric voltage. The controller calculates each difference of RMS value between the present and noncontrolled values. The present control type of the active harvester is decided from the sign and amplitude of the calculated RMS differences. The manipulated value of the threshold coefficient changes depending on the decision results corresponding to control types; this is to facilitate the active harvester to maintain the SCVS control type, which can provide a higher output voltage than other control types. The crisp control type decision leads to the unsteadiness of the manipulated value modification. The unsteadiness is due to the discontinuity of the manipulated value change, which is further caused by the discontinuity of the decision result change of the control type. Fuzzy control theory for the recognition of vague decisions is introduced to solve the discontinuity problems. It is possible that the proposed control strategy is robust to temperature fluctuations that may impact harvester dynamics. The strategy design is based on fuzzy control. Fuzzy control is used for robust control of systems that are difficult to model. The proposed strategy can incorporate small changes in dynamics due to temperature fluctuations into the modeling flexibility. The proposed control strategy is robust to vague decisions. Because the sensitivity of the strategy for vague decisions is difficult to investigate analytically, this sensitivity was confirmed by parametric simulations with respect to the relationship between the tuning parameters that link vague and concrete quantities and the output performance. In previous research, we confirmed that the active harvester with the proposed control strategy can exhibit higher output performance than the passive harvester without any tuning parameters.

## 3. Experiment

Figure 4 shows a model of the mechanical components of the harvester. The upper mass (Mass 1) hangs from the frame via a pantograph-shaped spring (Spring 1) that incorporates the piezoelectric transducer. The frame supporting the two vibrating masses does not vibrate; this is because the frame possesses sufficient rigidity and mass such that the vibrations are unable to transfer from the frame to the two vibrating masses and vice versa. The pantograph spring installing the cylindrical piezoelectric stack deforms elastically because of its infinitesimal deformation. This pantograph spring acts as a deformation reduction mechanism for the piezoelectric transducer. The vibration amplitude conversion rate by the pantograph spring is involved in the piezoelectric coefficient. Stiffness 1 includes the stiffness of the pantograph spring itself and the constant-charge stiffness of the transducer. The lower mass (Mass 2) is hung from the upper mass via a coil spring (Spring 2). The shaker connected to Mass 1 vibrates by the signal from the function generator. The masses were directly measured. The stiffness and damping coefficients were determined from the curve fitting of the structural frequency response. The frequency response between the input force and the first mass displacement of the 2-DOF vibrating structure under the open-circuit condition is shown in Figure 5. The constant-charge modal frequencies of the mechanical components were 18.8 and 35.0 Hz. The constant-voltage modal frequencies of the mechanical components (the modal frequencies for the closed circuit) were 18.7 and 34.5 Hz. Because the experimental harvester is fabricated so that its mechanical damping would be small, the anti-resonance at around 21.0 Hz may be clearly seen in the frequency response. The second mode is more damped than the first mode because of the plural components of the harvester. Because the experimental harvester is not equipped with an obvious dashpot, the dominant factor of each modal damping ratio is unclear, but it may come from material properties. This study aimed to evaluate strategies and not to develop a self-powered controller; therefore, fuzzy-based strategies were not implemented on the self-powered controller. The experimental controller driven by an external energy source outputs switch signals.

The piezoelectric coefficient was estimated from the slope of the line between the piezoelectric voltage and the first mass displacement in the phase plane. The slope, which is *b*_p_, was calculated from the least-squares estimation. The voltage was measured through the voltage follower to prevent the influence of the outflowing piezoelectric charge. The capacitance in the transducer, which is *C*_p_, was measured by the impedance analyzer. Because the deformation of the piezoelectric transducer during impedance analysis was minute, the measured capacitance was dealt with as the constant-strain capacitance. The inductor in the charge inversion circuit was selected to satisfy the frequency requirement under switching actions. The values of the piezoelectric and electric parameters denoted by *b*_p_, *C*_p_, *C*_s_, *L*, *R_L_*, and *R*_load_, were 4.5 × 10^5^ V/m, 4.3 × 10^−7^ F, 4.7 × 10^−5^ F, 2.0 × 10^−2^ H, 2.0 × 10^1^ Ω, and 2.2 × 10^6^ Ω, respectively.

The active fuzzy harvester with the ASCVS-t strategy was experimentally assessed under three disturbance conditions. The first disturbance condition provides a frequency sweep experiment from the first modal frequency to the second one. The mechanical displacement response from this disturbance condition is shown in Figure 6a. The small figures below the time history of the displacement show the power spectrum densities (PSDs) of the mechanical displacement in the first half time domain and the second half time domain. The disturbance has one dominant frequency; however, the frequency itself shifts from the first modal frequency to the second one. The second disturbance condition provides an experiment that changes the number of dominant frequencies from the unimodal to the double modal vibration of the harvester. The displacement response from this disturbance condition is shown in Figure 6b. From the figures of PSD in the second disturbance condition, the change in the number of dominant frequencies from one to two was confirmed. The third disturbance condition provides a random vibration experiment. This experiment was carried out by exciting the harvester with a random force through the shaker. The excitation signal driving the shaker is a band-limited noise produced by the function generator. Figure 6c presents the sample PSD of the mechanical displacement. Data in Figure 6 were measured under open-circuit conditions without any control. The harvesting experiments were implemented with identical disturbance histories.

The active harvester started switching actions 12 s after the start of the measurement. The initial value of the threshold coefficient of the ASCVS-t strategy was set to *β* [0] = 1.3 for all excitation conditions. For comparing the performance of the ASCVS-t strategy, the ASCVS strategy [37] and the LQR-based strategy [36] were adopted. They qualitatively evaluate inequality (6) using the displacement and piezoelectric charge estimated by the observer and do not consider inequality (6), respectively.

## 4. Results

The output voltage results (shown in Figure 7) implemented the STDH control from 0 to 12 s. The numerical results specified in the discussion were the amplification ratio of the output voltage at 80 s to the voltage in the range 0–12 s. The output voltages shown in Figure 7 under STDH control, which were the frequency sweep, the number of dominant frequency changes, and the random experiments, were 44.6, 45.0, and 51.3 V, respectively. The output voltage time histories with each switching strategy acquired in the disturbance frequency sweep experiment are shown in Figure 7a. Regardless of the strategy employed, the active harvester realized a higher output voltage caused by the switching action compared to STDH. The ASCVS and ASCVS-t strategies accomplished a higher output voltage than that of the LQR-based strategy because of the avoidance of undesirable switching actions. The converged output voltages with the ASCVS-t and ASCVS strategies, as well as with the LQR-based strategy, are 2.2 and 1.6 times higher than that of the STDH case, respectively. The ASCVS-t strategy had a faster time for the output voltage to converge to a steady state than the ASCVS strategy.

The time histories of the output voltages acquired in the experiment of the changing number of dominant frequencies are shown in Figure 7b. The trends of the converged output voltage magnitudes and charging speeds in each strategy were the same as in the sweep experiment results. The converged output voltages with the ASCVS-t and ASCVS strategies and with the LQR-based strategy are 2.1 and 1.3 times higher than that of the STDH case, respectively.

The time histories of the output voltages acquired in the random excitation experiment are shown in Figure 7c. The random experiment result was calculated from the average of the three trials. Remarkably, the order of the converged output voltage magnitude under the ASCVS-t and LQR-based strategies, and under the ASCVS strategy, differs from those of the other experiments. The ASCVS-t strategy provided a higher output voltage than that of the LQR-based one, similar to the previous results. Although the ASCVS strategy considers inequality (6), the output voltage of ASCVS is lower than those achieved by ASCVS-t and LQR-based strategies.

## 5. Discussion

The ASCVS-t strategy achieved a two times higher output voltage than the output voltage under the STDH control at 80 s in Figure 7a. The mechanical vibration at 80 s in Figure 7a was multimodal (see Figure 6a). Therefore, the ASCVS-t strategy is effective for the multimodal vibrating active harvester. In Figure 7a,b, the higher output voltages of both ASCVS and ASCVS-t strategies compared to the LQR-based switching strategy were introduced because of the desirable switching selection based on inequality (6). The output voltage in the ASCVS-t strategy case converged faster than that in the ASCVS strategy case. Fast charging was achieved by the tuning algorithm in the ASCVS-t strategy. The switching action is an important opportunity for active harvesters to amplify the piezoelectric charges and the corresponding output voltage. The number of switching actions in the ASCVS strategy case is smaller than that in the ASCVS-t strategy case because of the strict constraint related to the permission for executing switching actions. The ASCVS-t strategy did not continue performing SCVS control during the entire harvesting process; instead, it performed temporary SSHI control to boost the charging speed with consideration of the electromechanical harvester conditions. Although both the LQR-based and ASCVS strategies can execute only one control type—SSHI and SCVS, respectively—the ASCVS-t strategy is beneficial because it can achieve fast high-performance charging with switching control by continuously selecting two control types (SSHI and SCVS) owing to fuzzy control-based tuning. The decrease in output voltage due to the vibration suppression was clearly observed in the time histories from 12 to 15 s in Figure 7a,b. Because the LQR-based strategy did not consider inequality (6), the output voltage instantaneously increased at 12 s and gradually decreased. In the vibration environment shown in Figure 7a,b, the disturbance amplitude was constant prior to 40 s. The intermittent switching action always recovered the vibration amplitude suppressed by the switching action while maintaining the large vibration amplitude and corresponding output voltage. Although the disturbance frequency changed during the experiment, as shown in Figure 7a, the disturbance had only one dominant frequency during, before, and after the frequency changes. Therefore, both ASCVS and ASCVS-t strategies maintained the high output voltages owing to intermittent switching.

Figure 7b shows that the number of dominant frequencies of the disturbance increased after 40 s. Additionally, the number of mechanical vibration extremes per unit time increased. Both ASCVS and ASCVS-t strategies implement switching when the mechanical vibration reaches extreme values. Increasing the number of vibration extremes means that both strategies possess more opportunities for switching implementation. The difference between the ASCVS and ASCVS-t strategies is the constraint used for the switching implementation decision. The ASCVS strategy adopts the mathematically exact inequality (6) consisting of the state values at the present instant as the constraint; thus, the strategy had few switching actions per unit time due to the strict constraint. The ASCVS strategy result in Figure 7b shows the gradual increase in, and stagnation of, the output voltage repeatedly from 40 to 80 s. Appropriate switching actions increase the piezoelectric voltage amplitude and the corresponding output voltage. The time history of the output voltage shows that the ASCVS strategy did indeed provide an appropriate switching action. However, the implementation of switching was discontinuous, and its operation was unstable. The constraint of the ASCVS-t strategy was a threshold derived from the global variation of the state values in the harvester, which mitigated the switching implementation constraint compared to inequality (6). The trends of the state values were incorporated into the RMS calculation. The output voltage amplification of the ASCVS-t strategy was implemented faster than that of the ASCVS strategy owing to mitigation constraints and the trend consideration feature of the ASCVS-t strategy.

When controlling for random dynamics (Figure 7c), strategies must consider the expected value or trend of the random dynamics instead of the instant state values. Compared to the ASCVS-t and both LQR-based and ASCVS strategies, the major difference in each control depends on the time range of the considerations of the dynamics trend. The ASCVS strategy considers only the present harvester conditions for the decision of the switching action executions; both ASCVS-t and LQR-based strategies consider the change in the harvester conditions over a wide time range. Because the ASCVS-t strategy calculates the manipulated value of the threshold coefficient from changes in the electromechanical conditions with the RMS calculation, this strategy can reflect the trend of the random dynamics of the harvester learned from the past state information of the harvester for the switching actions. The LQR-based strategy uses the LQR gains calculated with an evaluation index that considers the future of infinity as the upper limit of the integration range. That is, this strategy can also reflect the trend of the random dynamics learned from the future state information for the switching actions. In contrast, the ASCVS strategy uses only the present electromechanical values of the harvester and decides whether a present switching action is desirable. The trend of random dynamics does not reflect switching execution. The ASCVS strategy implements the selection of switching actions only from a microscopic perspective. Therefore, the number of switching actions was significantly lower than that of the other control strategies. The output voltage did not become sufficiently large because the piezoelectric charge was not amplified when no switching action was conducted. The ASCVS-t strategy maintained the high output voltage amplitude due to its trend consideration feature. In the random vibration environment, pausing the switching action does not always provide vibration recovery because the future disturbance is unknown. Therefore, the ASCVS strategy, which determines the implementation of the switching action based on the present time conditions, could not achieve a sufficiently high output voltage. The ASCVS-t strategy, which considers the trend of the state values, solved the problem posed by the ASCVS strategy and thus maintained the required output voltage amplitude.

Robustness discussed in this research refers to the ability to maintain a high output voltage without any off-line works such as the manual interference adjusting parameters and gains in the control strategy despite the harvester being subjected to various disturbances. The harsh environment against the control strategies includes the unknown disturbance exposures to the harvester. Deriving an analytical solution of the harvesting performance guarantees that a control strategy will improve the performance. In cases wherein it is difficult to perform disturbance modeling in advance, derivation of an analytical solution of the harvesting performance is equally difficult. In particular, active harvesters with charge inversion circuits pose added difficulty in the derivation of analytical solutions than that of the passive harvesters; this is due to the nonlinear electromechanical coupling dynamics in active harvesters. To demonstrate the robustness of the control strategy, operations of the harvester with the proposed control strategy under various disturbance environments are necessary. The three experiments in Figure 7 show the operation of the control strategy with identical parameters and gains for three different disturbances. In all the experiments, the proposed control strategy maintained a high output voltage. Therefore, the proposed ASCVS-t strategy is robust.

The output voltage obtained from the experimental apparatus can accommodate the power required to operate the controller. Details of the power consumption of the self-powered controller have been reported in our previous publications. The self-powered controller was operated within a power consumption range of 1.10–13.7 mW [34,36]. The variation in the power consumption of the self-powered controller depended on the calculation amount of installed control strategies and the number of driving A/D converters. The ASCVS-t strategy implementation with the fuzzy control theory may require large power consumption owing to complex calculations. The power consumption of the ASCVS-t strategy implementation is estimated as 27.4 mW; this is two times higher than the worst power consumption achieved in the previous reports. The energy consumption per second is 27.4 mJ. The harvesting experiment results shown in Figure 7 achieved approximately 45.0 V output under STDH control. When a smoothing capacitor is regarded as an apparent battery, the stored energy in the battery is obtained as
(11)$$E_{\mathrm{harvest}}=\frac{1}{2}C_{s}V_{\mathrm{out}}^{2}=47.6\;\left(\mathrm{mJ}\right)$$

The stored energy in the battery is greater than the required energy of the self-powered controller; consequently, the controller can accomplish self-powered operation only when the battery stored sufficient energy. The purpose of this paper is not to discuss the self-powered operation but the operational performance of the ASCVS-t strategy under several disturbance conditions. The implementation will be presented in the next paper that we are preparing.

The effectiveness of the vibration energy harvesters equipped with nonlinear mechanical components has been discussed [55,56,57,58]. They reported that the initial values determine their superiority or inferiority of performance between harvesters equipped with linear and nonlinear mechanical components. Because the ASCVS-t strategy adapts to the active harvester housed linear mechanical components, the harvesting performance of this harvester does not depend on the initial values.

## 6. Conclusions

In this paper, the ASCVS-t switching strategy demonstrated a high output voltage from an active harvester with the charge inversion circuit. It was assessed under nonstationary and random vibration conditions as realistic vibrations. The ASCVS-t strategy could select the switching actions that amplify the output voltage with the threshold as the assessment criterion for the piezoelectric voltage. Because the threshold used in the ASCVS-t strategy was time varying considering the harvester conditions, the ASCVS-t strategy can maintain an efficient switching action even if the disturbance is nonstationary and random during the harvesting process. The thresholds adjust the intervals between switching actions to reflect changes in the piezoelectric voltage and mechanical displacement caused by the switching action itself, thereby forcing a nonstationary disturbance. Owing to the fuzzy control theory adopted as the tuning algorithm for the threshold coefficients, the changes in both electromechanical variables can be treated ambiguously. The ambiguous criterion realizes robust tuning even under unknown disturbance conditions.

The experiments confirmed that the ASCVS-t strategy maintains a high output voltage when the harvester was exposed to nonstationary disturbances. The comparison of the magnitude of the output voltages with two previous switching strategies under each excitation experiment confirmed that the ASCVS-t strategy properly implemented efficient harvesting because of the selection of the desirable switching actions with the tuned time-varying threshold.

The proposed control strategy can increase both output voltage and energy. Therefore, the proposed strategy is suitable for an operational scenario that alternately implements harvesting and consumption. As an application area, the proposed control strategy can be used to power a tire-pressure monitoring system. The implementation of the proposed strategy needs sensors measuring the voltage and vibration, and a controller equipped with high computational power. The self-powering of these devices is a technical issue of the implementation. Our self-sensing technique and the self-powered digital controller for a harvester can solve the above technical issue.

In this paper, the ASCVS-t strategy was implemented in a powered active harvester controlled by a PC with an external power source. The implementation of the ASCVS-t strategy in a self-powered active harvester is the next research goal.

## Figures and Tables

**Figure 1 sensors-21-03913-f001:**
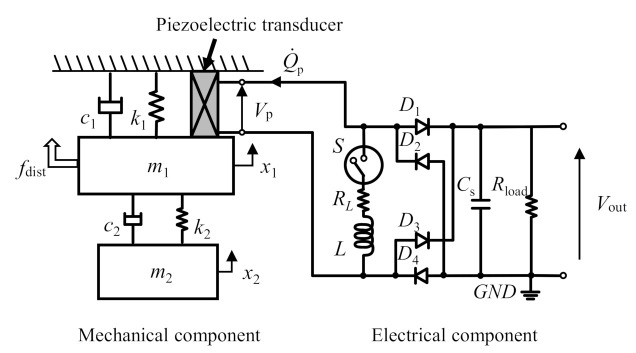
Schematic of the piezoelectric vibration energy harvester composed of a piezoelectric transducer, 2-DOF vibration structure, and harvesting circuit with a charge inversion circuit.

**Figure 2 sensors-21-03913-f002:**
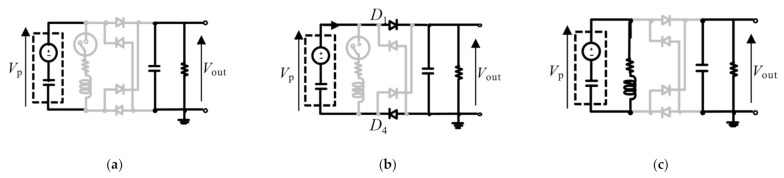
Circuit connections under each circuit equation. The black and gray lines show the conductive and nonconductive lines. (**a**) The piezoelectric charge is constant because of the nonconductive conditions of the switch and rectifier. (**b**) The piezoelectric charge outflows from clamped storage in the piezoelectric transducer to the smoothing capacitor owing to the conductive rectifier. This diagram indicates the condition in which the piezoelectric voltage sign is positive. (**c**) The piezoelectric charge vibrates through the inductor and capacitor. This connection is maintained for half of the *LC* resonance frequency time.

**Figure 3 sensors-21-03913-f003:**
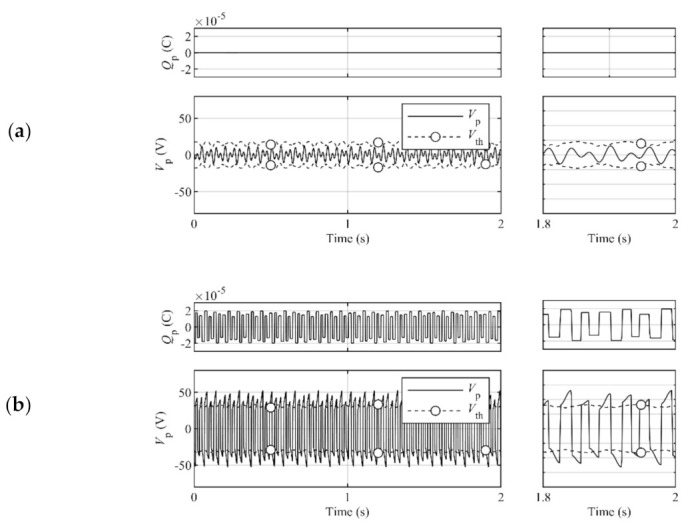
Time histories of the piezoelectric charge, and the piezoelectric voltage and the threshold under each control type. (**a**) The STDH control type, which occurs because of the exceedingly large threshold coefficient magnitude, does not perform switching actions. Although the charge moves slightly in accordance with the harvesting circuit state, the change in the charge under STDH conditions is hardly visible because the scale of the vertical axis is matched with the other control types. (**b**) The SSHI control type, which occurs because of the exceedingly small threshold coefficient magnitude, performs switching actions at all peaks. (**c**) The SCVS control type, which occurs because of the adequate threshold coefficient magnitude, accomplishes appropriate intermittent switching actions.

**Figure 4 sensors-21-03913-f004:**
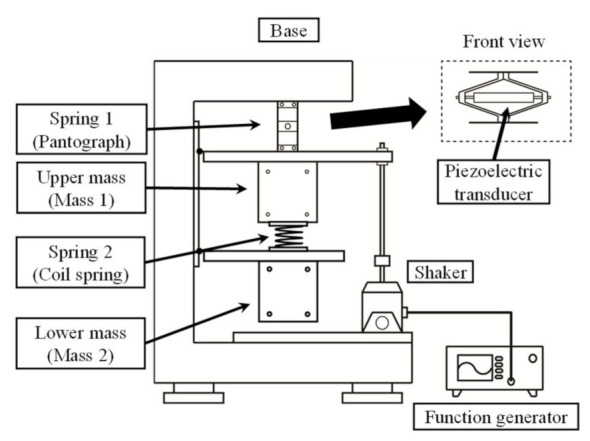
Experimental apparatus for assessment experiments.

**Figure 5 sensors-21-03913-f005:**
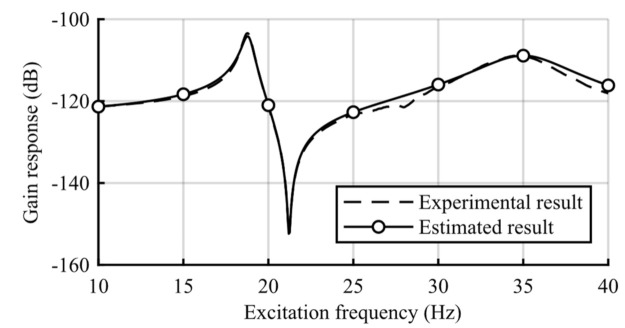
The frequency response between the input force and the first mass displacement of the 2-DOF vibrating structure under the open-circuit condition.

**Figure 6 sensors-21-03913-f006:**
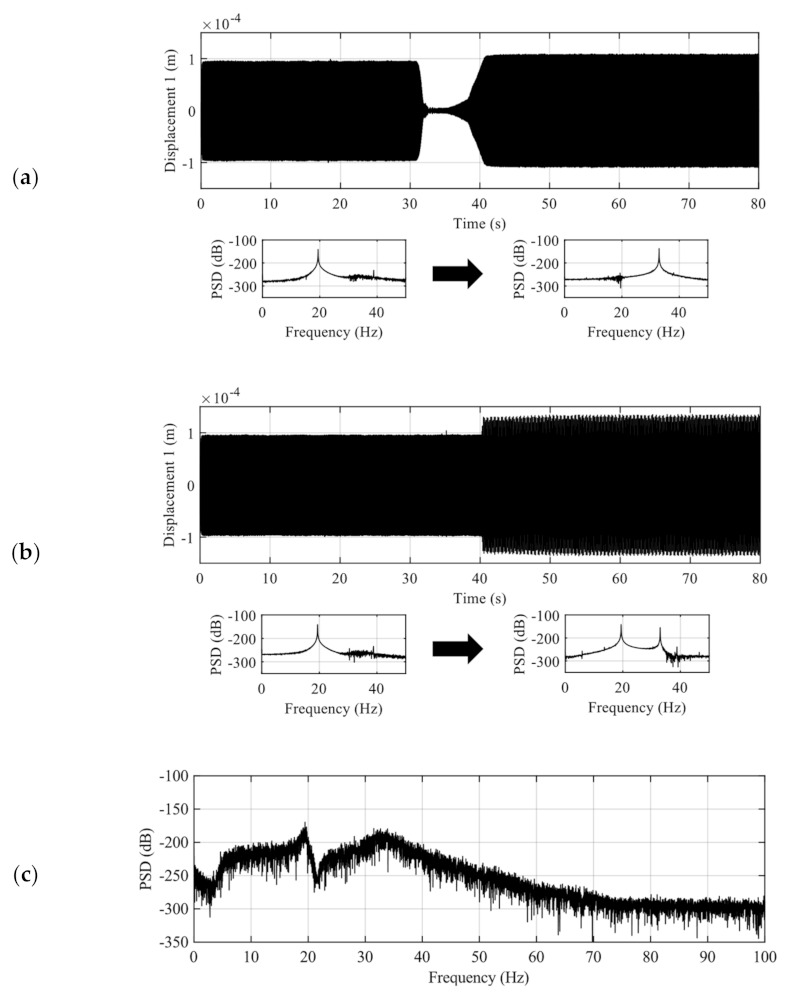
Three nonstationary and random disturbance conditions in the experiments. (**a**) Frequency sweep vibration experiment: the upper figure shows the time history of displacement 1; the lower figures show each PSD in response to vibration changes. (**b**) Change in the number of dominant frequency vibration experiment: each figure shows the same information as (**a**). (**c**) Random vibration experiment: sample PSD is shown.

**Figure 7 sensors-21-03913-f007:**
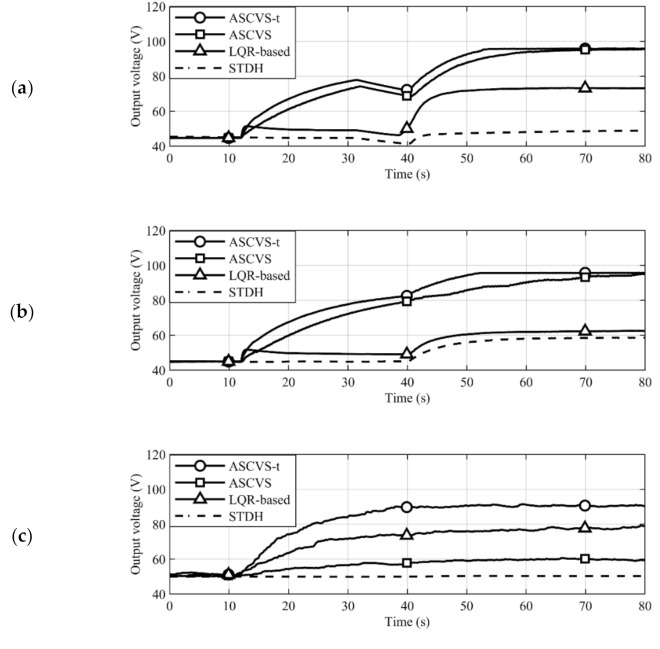
Time histories of output with each switching strategy. (**a**) Frequency sweep experiment. (**b**) Experiment on change in the number of dominant frequencies. (**c**) Random experiment.

## Data Availability

Any data that support the finding of this study are included within the article.

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
