# Peer review of "Adaptive and Robust Operation with Active Fuzzy Harvester under Nonstationary and Random Disturbance Conditions"

_sensors, 2021, doi:10.3390/s21113913_

Round 1
Reviewer 1 Report
Report on paper "Adaptive and robust operation with active fuzzy harvester under nonstationary and random disturbance conditions" submitted by Hara et al., for publication in Sensors (sensors-1227805).
The authors developed an adaptive switching considering vibration suppression-threshold strategy. They evaluated the proposed strategy in term of output performances under three realistic vibration conditions and performed experimental validation. While the paper is interesting, it cannot be accepted in its present form and the authors must perform some modifications by addressing the following comments:
1- In figure 5, the authors could explain/justify the anti-resonance at around 21 Hz. Why the second mode is more damped than the the first mode?
2- In the experiments, the constant-charge modal frequencies of the mechanical components are separated in frequency. What about the case of closed modes?
3- The role of damping and its influence on the stability of the proposed control strategy should be discussed.
4- What about the sensitivity of the proposed strategy to temperature fluctuations that may impact the harvester dynamics?
5- In the conclusion, the limitations of the proposed strategy should be specified from a critical point of view.
Reviewer 2 Report
The manuscript is detailed and generally well-written. The different sections and subsections follow each other logically and give a good flow of information throughout the text. The manuscript systematically introduced a switching strategy to solve problem of time-invariant threshold under nonstationary vibration conditions.This work is very valuable for research in the application field of piezoelectricvibration energy harvester.however, below are important points that need to be addressed with minor comments on room for improvement:
- the vague decision of the ASCVS-t strategy can alternately change thecontrol style between intermittent and continuous control depending on the disturbance and theharvester conditions. But, how sensitive is the system via the vague decision of the ASCVS-t strategy.
- In your article, what is the field of application of this ASCVS-t strategy and its limits.
- The language is generally good, but there are some sentences which should be improved and some written errors which should be made corrections.
Round 2
Reviewer 1 Report
The authors have addressed my comments sufficiently to recommend publication of the paper in its current form.
This manuscript is a resubmission of an earlier submission. The following is a list of the peer review reports and author responses from that submission.
Round 1
Reviewer 1 Report
Review of the manuscript sensors-1061736
The manuscript evaluates the ASCVS-t strategy for the active piezoelectric energy harvester under three realistic vibration conditions. This strategy implements a tuning algorithm for the time-varying threshold and intermittent switching employing the fuzzy control theory. The results confirmed that the presented strategy achieves a greater output voltage than the existing strategies under all nonstationary vibration conditions.
The manuscript is well written. The proposed experimental evaluation of the strategy is explained and discussed appropriately.
In my opinion, the manuscript can be accepted after clarification of the several issues:
1. Lines 268-269: Considering that the manuscript proves the effectiveness of the ASCVS-t strategy, its short description would be of interest to readers.
2. Lines 300-302: The sentence is a little bit confusing. Please consider rewriting it.
3. Lines 350-351: I believe, according to Fig. 8(c), that ASCVS-t and ASCVS are swapped in this sentence.
Reviewer 2 Report
See attached pdf

Reviewer 3 Report
Report on paper "Adaptive and robust operation with active fuzzy harvester under nonstationary and random disturbance conditions" submitted by Hara et al., for publication in Sensors (sensors-1061736).
The authors developed an adaptive switching considering vibration suppression-threshold strategy. They evaluated the proposed strategy in term of output performances under three realistic vibration conditions and performed experimental validation. Although the paper is interesting and includes experiments to support the simulated models, it cannot be accepted in its present form and the authors must perform some modifications by addressing the following comments:
- In the abstract, the authors could quantify the performances of the proposed strategy.
- In the introduction, the literature survey lacks of references in the field of vibration energy harvesting under random vibrations, which is a topic deeply investigated in the recent past (for instance (i) [Sensors 2020, 20(19), 5456], (ii) [Sensors and Actuators A Physical, 283, 54-64, 2018]).
- In the introduction, the originality of the paper should be clearly highlighted with respect to the previous authors publications and any overlap with respect to ref 56 and 57 should be reduced.
- In section 2, the authors should specify and justify the assumptions used for the electromechanical model.
- Sections 4 and 5 should be extended where some discussions lack of depth and the authors should provide a detailed explanation of each result according to the physics phenomena and the significance of each result for energy harvesting.
- What about the robustness of the proposed strategy against harsh environments? This point should be discussed.
Round 2
Reviewer 2 Report
Revisions have satisfactorily addressed minor issues with clarity and presentation.
Revisions did not satisfactorily address
1. The literature review which still contains inappropriate self-citation in place of more impactful works in the literature
2. the overall scope of the article which is too narrow and would do better to combine with future work.
Reviewer 3 Report
While the first revision improved partially the paper quality, some important points must be addressed:
1- The role of damping and its influence on the stability of the proposed control strategy should be discussed.
2- What about the effectiveness of the proposed control strategy in the case of nonlinear and multimodal vibration energy harvesters ((i)[Journal of Sound and Vibration, 319, 515–30, 2009], (ii) [Smart Materials and Structures, 28, 07LT02, 2019], (iii) [Journal of Sound and Vibration, 329, Issue 9, 1215-1226, 2010], (vi) [Smart Materials and Structures, 29, 10LT01, 2020])? This important point should be discussed.
3- In the experiments, the constant-charge modal frequencies of the mechanical components are separated in frequency. What about the case of closed modes?
4- The number of auto-citations is extremely high with respect to the total number of references (27/61). I do not think that all these citations are justified and I invite the authors to reduce this ratio significantly and include more relevant references from the literature.
5- The quality of figures 6 and 7 should be enhanced.
